# Impact of COVID-19 pandemic on breast and cervical cancer screening in Denmark: A register-based study

Mette Hartmann Nonboe[1]*, George Napolitano[2], Jeppe Bennekou Schroll[3], Ilse Vejborg[4], Marianne Waldstrøm[5], Elsebeth Lynge[1]

[1]Center for Epidemiological Research, Nykøbing Falster Hospital, Nykøbing Falster, Denmark; [2]Department of Public Health, University of Copenhagen, Copenhagen, Denmark; [3]Department of Gynecology and Obstetrics, Odense University Hospital, Odense, Denmark; [4]Department of Breast Examinations, Copenhagen University Hospital Herlev/Gentofte, Copenhagen, Denmark; [5]Department of Pathology, Aarhus University Hospital, Aarhus N, Denmark

## Abstract

**Background:** Denmark was one of the few countries where it was politically decided to continue cancer screening during the COVID-19 pandemic. We assessed the actual population uptake of mammography and cervical screening during this period.

**Methods:** The first COVID-19 lockdown in Denmark was announced on 11 March 2020. To investigate possible changes in cancer screening activity due to the COVID-19 pandemic, we analysed data from the beginning of 2017 until the end of 2021. A time series analysis was carried out to discover possible trends and outliers in the screening activities in the period 2017–2021. Data on mammography screening and cervical screening were retrieved from governmental pandemic-specific monitoring of health care activities.

**Results:** A brief drop was seen in screening activity right after the first COVID-19 lockdown, but the activity quickly returned to its previous level. A short-term deficit of 43% [CI –49 to –37] was found for mammography screening. A short-term deficit of 62% [CI –65 to –58] was found for cervical screening. Furthermore, a slight, statistically significant downward trend in cervical screening from 2018 to 2021 was probably unrelated to the pandemic. Other changes, for example, a marked drop in mammography screening towards the end of 2021, also seem unrelated to the pandemic.

**Conclusions:** Denmark continued cancer screening during the pandemic, but following the first lockdown a temporary drop was seen in breast and cervical screening activity.

**Funding:** Region Zealand (R22-A597).

## Editor's evaluation

Denmark was one of the few countries that did not suspend cancer screening in the early stage of the COVID-19 pandemic. This important study offers convincing evidence of how the pandemic impacted the use of breast and cervical cancer screening services. This article has a broad interest to public health researchers and health policy implementation.

*For correspondence: menon@regionsjaelland.dk

## Introduction

For almost 2 years, COVID-19 has been a central part of daily life in Denmark. On 11 March 2020, when the WHO declared COVID-19 a global pandemic (*WHO, 2020*), the prime minister of Denmark announced a national lockdown (*Danish Government, 2020*). In many ways, the first lockdown in Denmark was similar to the handling of the pandemic in other countries with, among other restrictions, temporary suspension of non-urgent health care services. However, as opposed to the situation in, for example, the UK and Italy, the cancer screening programmes were not put on hold (*Danish Health Authority, 2022a*; *Maringe et al., 2020*; *Vanni et al., 2020*). By 22 January 2022, the last COVID-19 restrictions in Denmark were repealed.

Cancer screening in Denmark is free of charge, so even for citizens economically affected by COVID-19, payment should be no hindrance to participation. However, at the lockdown press conference, the prime minister and the head of the Danish Health Authority focused on the need to protect the health care system by avoiding unnecessary contact with general practitioners (GPs) and emergency departments. Some people could have interpreted this as a 'stay home' message from cancer screening as well (*Wilson et al., 2021*). The screening programmes could also have been affected by, for example, relocation of personnel to COVID-19 tasks and use of HPV-testing equipment for analysis of COVID-19 tests. For individual citizens existing barriers to participation in cancer screening could also have been enhanced during the pandemic, for example, difficulties in booking consultations at their GPs (*Wilson et al., 2021*), and new barriers could have arisen, such as concern about becoming infected with COVID-19 (*Wilson et al., 2021*); concern about unnecessary burdening of the health care system (*Wilson et al., 2021*); or simply by changes of priorities (*Castanon et al., 2021*). In the present study, we assessed the effect of the COVID-19 pandemic on the number of tests performed in mammography and cervical screening, nationally and by region.

## Materials and methods

### Setting

Health care in Denmark is financed through taxes, and cancer screening, assessment, and treatment of detected lesions are free of charge for the patient. Denmark has three national cancer screening programmes. First, biennial mammography screening for women aged 50–69 occurs at specialised hospital clinics (*Danish Health Authority, 2019a*). It is a stand-alone programme including only mammography screening taken in the public sector after invitation. In Denmark, opportunistic breast screening is minimal (*Jensen et al., 2005*). Second, third or fifth yearly cervical screening for women aged 23–64, where samples are collected primarily in the private sector by GPs or office-based gynaecologists (*Danish Health Data Authority, 2020*). It is an integrated programme including both cell samples collected after invitation, cell samples collected at the woman's/doctor's initiative, or collected as a control of a previous abnormal test. A unique screening initiative took place in 2017, where women above 70 had a one-time invitation to screening (*Region Midtjylland, 2018*), as this birth cohort has not previously been offered screening. Lastly, biennially screening for bowel cancer. As this programme is based on faecal immunological test kits sent to people's home, the programme was not part of the Danish health care monitoring system from which we extracted data (*Danish Health Authority, 2019b*). The five Danish regions are responsible for the programmes, following the Danish Health Authority guidelines.

Health care activities are registered centrally. Based on these registered data, the Danish Health Authority closely monitored health care activities in hospitals and the private practice sector during the pandemic. From 10 June 2020 and up until 5 January 2022, activity data were published regularly in 13 reports (*Danish Health Authority, 2022b*). The present study was based on the activity data collected in this monitoring effort.

At mammography screening, two views are taken, a craniocaudal and a mediolateral oblique. The screens are read independently by two radiologists, now in one region by one radiologist and an AI-based system, and if inconsistent, a consensus decision is reached. The screening result is reported as normal or abnormal; in the latter case, the letter is combined with an invitation to clinical assessment. The result is communicated to the woman via her Digital Post for communication with public authorities. For simplicity, we used the term 'mammography screen' for the entire mammography examination.

At cervical screening, a cell sample is taken by the GP/gynaecologist, placed in a vial, and sent to the regional pathology laboratory for cytology examination, now also primary HPV test in an implementation trial. The result is communicated both to the woman and to the cell-sample collector. In cases of milder abnormalities, the woman is recommended cell-sample control. For more severe abnormalities, the woman is referred to a gynaecologist.

At the start of the lockdown on 13 March 2020, emphasis was, among other things, put on limiting the spread of COVID-19 to health care staff. For the GPs, the Organisation of General Practitioners in Danish called Praktiserende Lægers Organisation, PLO, specified guidelines dividing patient contacts into those needing consultation and those that could be handled by phone/web or could eventually be postponed (*DSAM, 2022a*). In the first GP guideline from 25 March 2020, cervical screening was recommended to be postponed. However, the Danish Health Authority intervened, and 9 days later, on 3 April 2020, a new GP guideline stated that cervical screening was not suspended (*DSAM, 2022b*). Furthermore, to compensate for the observed drop in mammography screening in the early phase of the pandemic, all five regions issued extra reminders to women who had cancelled their screening appointments in June and July 2020 (*Danish Cancer Society, 2020a*).

Throughout the COVID-19 lockdown, the Danish Cancer Society and news agencies regularly reported on the state of cancer-related activities, including trends in numbers of mammography screens and cell samples (*Danish Cancer Society, 2020b*; *Danish Health Authority, 2022b*). The Danish Cancer Society encouraged women who had skipped cervical screening to book a new appointment with their GP (*Danish Cancer Society, 2020b*).

## Data

For mammography screening, the government retrieved from the National Patient Register (Landspatientsregistret [LPR]) using the activity codes DZ123A (mammography screening), DZ123AA (mammography screening under the Health Act), and DZ108A (now expired code), and the procedure code UXRC45 (mammography, screening) alone or combined with additional diagnosis codes ZPROON (normal test result) and ZPR01N (abnormal test result). The same data source and codes are used in the annual reporting from the Danish Quality Database on Mammography Screening (*DKMS, 2021*). Data on cervical screening were retrieved from the Danish Health Insurance Register using the combination of provider codes 80 (GP) and 07 (office-based gynaecologist) with payment codes 2102 and 4301 (cervical cell sample). The data source and codes differ from the ones used in the annual reporting from the Danish Quality Database on Cervical Cancer Screening (*DKLS, 2022*) where data are retrieved from the National Pathology Register. We retrieved data from 2017 to 2021, with the period before 13 March 2020 representing the pre-COVID-19 period.

All data were available in weekly aggregated form, using a standard ISO-week numbering system. The Danish Health Data Authority provided mammography data for 2017–2021. Cell-sample data for 2017–2019 and from weeks 34 to 52 in 2021 were provided by the Danish Health Data Authority, while we extracted cell-sample data from 2020 and weeks 1 to 33 in 2021 from eSundhed.dk (*Danish Health Data Authority, 2022*).

## Data analysis

To identify a possible impact of the COVID-19 epidemic on cancer screening activity, we undertook a time series analysis. To easily identify seasonal patterns, especially moving holidays, for example, Easter, summer holidays, etc., data were split into periods of 4 weeks, called 'months' in the rest of this paper, making a total of 13 months per year, with week 53 of 2020, a leap year, included in month 13 (*Supplementary file 1*).

Seasonal and trend patterns were then identified through a Seasonal Decomposition of Time Series by Loess (STL decomposition) with robust fitting. The seasonal-adjusted time series, that is the original time series without the seasonal component, were then modelled with a negative binomial integer-valued GARCH model with a logarithmic link (*Liboschik et al., 2017*). Outliers and linear trends in the time series were identified through graphical inspection and analytical procedure. In particular, outliers were identified through the procedures described in *Liboschik et al., 2017*; here, outliners are named 'interventions'and in *López-de-Lacalle, 2019* using the 'tso function' in the tsoutlier package. Linear trends were found by searching for a statistically significant linear term in the regression model. Once a liner trend was identified, the start date was chosen as the date minimising the AIC of the model.

Estimated reductions in the number of screens were obtained from the coefficients of the relevant 'intervention' covariate of the model. The corresponding 95% confidence intervals are calculated from the coefficients' standard error using normal approximation.

All analyses and plots were done using R ver. 4.1.1 (*R Development Core Team, 2021*), with the package collection tidyverse (*Wickham et al., 2019*) and the packages tscount (*Liboschik et al., 2017*) and tsoutlier (*López-de-Lacalle, 2019*).

## Results

### Mammography screening

On average, 287,284 mammography screenings were performed annually from 2017 to 2019, compared with 260,508 in 2020 and 270,303 in 2021 (*Table 1*).

The time trend analysis for mammography screening is illustrated in *Figures 1–3*. *Figure 1* depicts the observed weekly number of mammography screening. *Figure 2* shows the decomposition of the monthly data. In the seasonal component, one can clearly identify the prominent low peaks due to Easter, month 4, summer, month 8, and Christmas, month 13. In *Figure 3*, the fitted model is depicted together with the seasonal-adjusted data.

We found that from 24 February 2020 to 22 March 2020, there was a –22.6% [CI –30.1 to –14.4] reduction in the number of mammography screenings, and from 23 March 2020 to 19 April 2020, a –43.3% [CI –49.0 to –36.9] reduction (*Figure 3* and *Table 2*). The activity decline was short-termed; however, it did not reach the previous levels in the first months after the first lockdown. There was a statistically significant decrease of –17.0% [CI –21.2 to –12.6] in the number of mammography screens from 16 August 2021 to 2 January 2022 (*Figure 3*). All percentages were with respect to the expected number in the absence of the outlier. Separating the numbers by region, the Capital Region and the much smaller Region North were responsible for the drop at the end of 2021 (*Figure 3—figure supplement 1*). In the Capital Region, from 19 June 2021 to 15 August 2021, the decrease was –34.0% [CI –43.4 to –23.1]; from 16 August 2021 to 12 September 2021 –67.2% [CI –72.1 to –61.3]; from 13 September 2021 to 10 October 2021 –85.3% [CI –88.1 to –81.9], and from 11 October 2021 to 2 January 2022 –41.3% [CI –49.5 to –31.8].

**Table 1.** Number of mammography and cervical screens by year and region, Denmark 2017–2021.

| Region | Mammography screening | | | | | Cervical screening | | | | |
|---|---|---|---|---|---|---|---|---|---|---|
| | Year 2017 | Year 2018 | Year 2019 | Year 2020 | Year 2021 | Year 2017 | Year 2018 | Year 2019 | Year 2020 | Year 2021 |
| Denmark | 289,150 | 290,275 | 282,426 | 260,508 | 270,303 | 443.126 | 348,950 | 344,045 | 306,472 | 323,598 |
| Capital | 78,323 | 75,415 | 75,964 | 65,012 | 51,457 | 155,022 | 124,149 | 120,329 | 110,655 | 116,145 |
| Central | 67,282 | 68,117 | 66,895 | 63,043 | 69,719 | 97,775 | 77,200 | 84,099 | 72,006 | 75,967 |
| North | 30,835 | 31,033 | 30,415 | 31,978 | 26,532 | 43,818 | 34,435 | 33,719 | 29,293 | 32,288 |
| South | 70,396 | 69,468 | 68,271 | 64,108 | 72,876 | 88,540 | 67,896 | 64,972 | 57,108 | 59,738 |
| Zealand | 42,314 | 46,242 | 40,881 | 36,367 | 49,719 | 57,971 | 45,270 | 40,926 | 37,410 | 39,460 |

Source: Own calculations based on numbers provided by the Danish Health Data Authority.

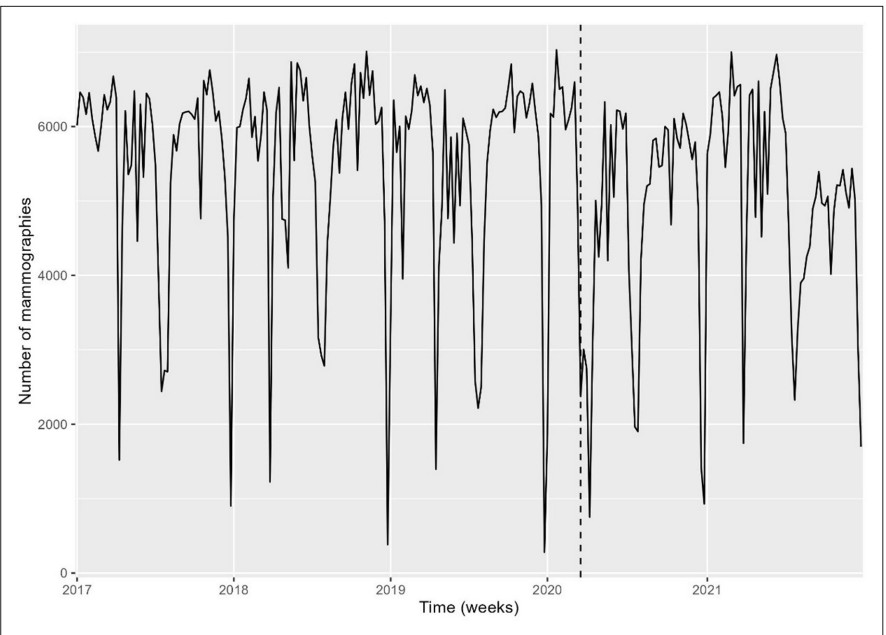

**Figure 1.** Weekly number of mammography screens, Denmark 2017–2021. Source: Own calculations based on numbers provided by The Danish Health Data Authority.

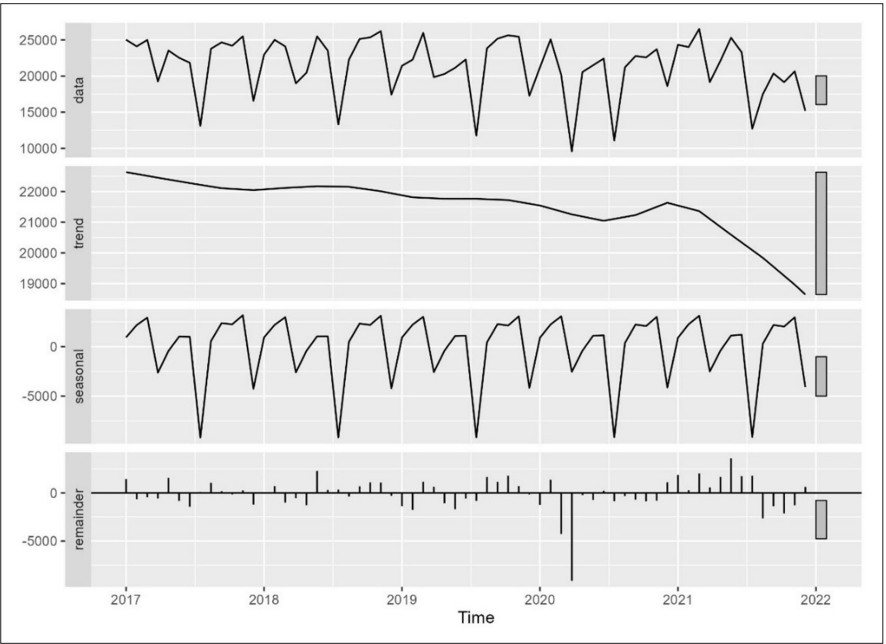

**Figure 2.** Decomposition of time series in number of mammography screens into seasonal, trend, and irregular component, Denmark 2017–2021. Source: Own calculations based on numbers provided by The Danish Health Data Authority.

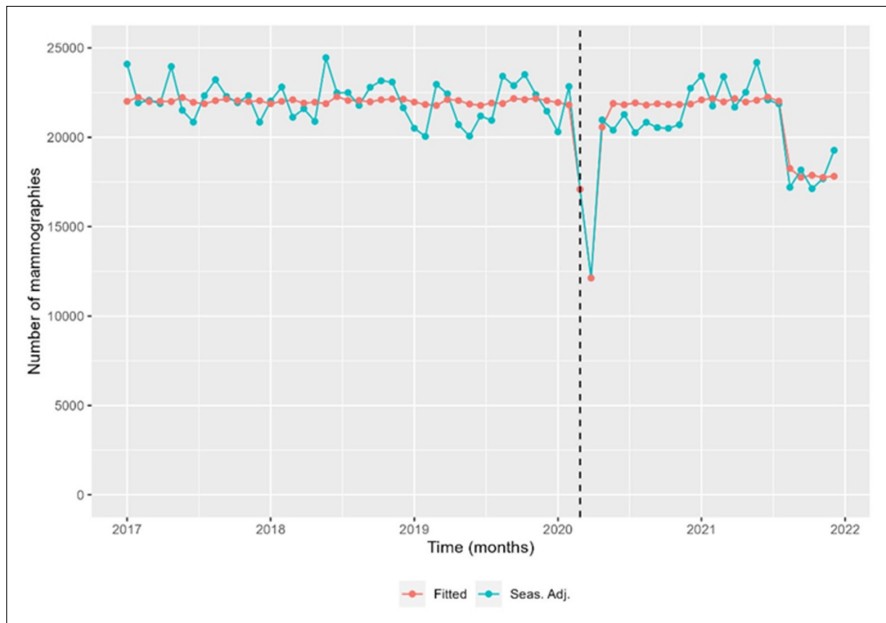

**Figure 3.** Seasonal-adjusted time series and fitted model of the number of mammography screens, Denmark 2017–2021. Source: Own calculations based on numbers provided by The Danish Health Data Authority.

The online version of this article includes the following figure supplement(s) for figure 3:

**Figure supplement 1.** Seasonal-adjusted time series and fitted model of the number of mammography screens, by region 2017–2021.

**Table 2.** Estimated changes in number of seasonally adjusted mammography screens.
Denmark and regions, 2017–2021. Reductions and increases are with respect to the average base line.

|  | Time period | Estimated change in percent |
|---|---|---|
| Denmark | 24 February 2020 – 22 March 2020<br>23 March – 19 April 2020<br>16 August 2021 – 02 January 2022 | –22.6% [CI –30.1 to –14.4]<br>–43.3% [CI –49.0 to –36.9]<br>–17.0% [CI –21.2 to –12.6] |
| Capital | 30 December 2019 – 18 July 2021<br>23 March 2020 – 19 April 2020<br>19 July 2021 – 15 August 2021<br>16 August 2021 – 12 September 2021<br>13 September 2021 – 10 October 2021<br>11 November 2021 – 02 January 2022 | –11.8% [CI –15.7 to –7.7] – (level shift)<br>–43.5% [CI –51.6 to –34.1]<br>–34.0% [CI –43.4 to –23.1]<br>–67.2 [CI –72.1 to –61.3]<br>–85.3 [CI –88.1 to –81.9]<br>–41.3 [CI –49.5 to –31.8] |
| Central | 24 February 2020 – 22 March 2020<br>23 March 2020 – 19 April 2020 | –15.7% [CI –24.4 to –6.1]<br>–38.2% [CI –44.7 to –30.9] |
| North | 24 May 2021 – 02 January 2022 | 7.5% [CI 5.1–9.9] |
| South | 23 March 2020 – 19 April 2020 | –43.5% [CI –53.5 to –31.4] |
| Zealand | 24 February 2020 – 19 April 2020<br>02 November 2020 – 02 January 2022 | –51.5% [CI –58.5 to –43.3]<br>1.6% [CI 0.7–2.5] |

Source: Own calculations based on numbers provided by The Danish Health Data Authority.

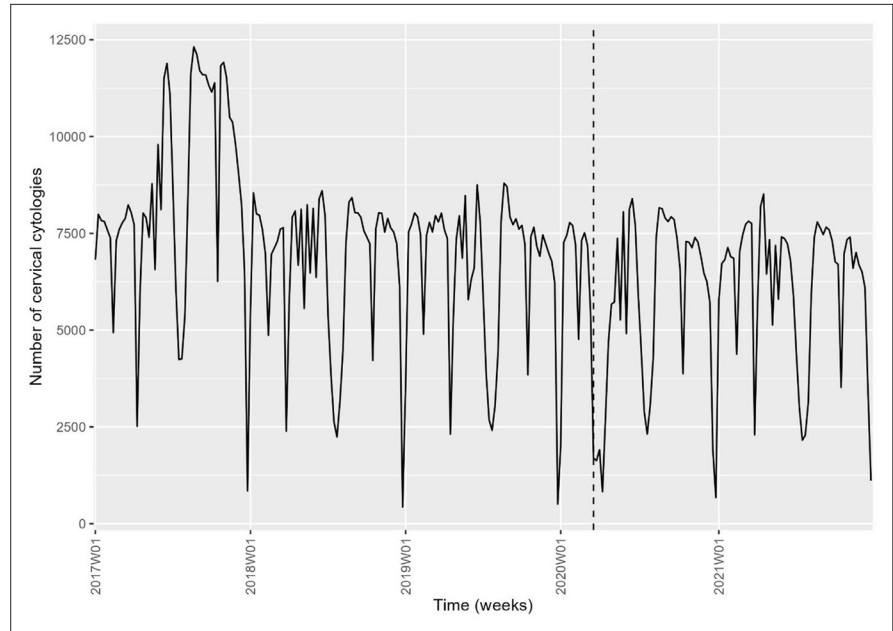

**Figure 4.** Time series of the weekly number of cervical, Denmark 2017–2021. Source: Own calculations based on numbers provided by The Danish Health Data Authority.

## Cervical screening

On average, 346,498 cell samples were collected annually from 2018 to 2019, compared with 306,472 in 2020 and 325,598 in 2021 (**Table 1**).

The time series analysis for cervical screening is illustrated in **Figures 4–6**. Analysis was done the same way for mammography screening analysis. Overall, there was an annual, statistically significant reduction of –1.9% [CI –2.9 to –0.8] from 6 November 2017 to 2 January 2022 in the number of cell samples in Denmark (**Table 3**). The special initiative for screening elderly women was issued in 2017,

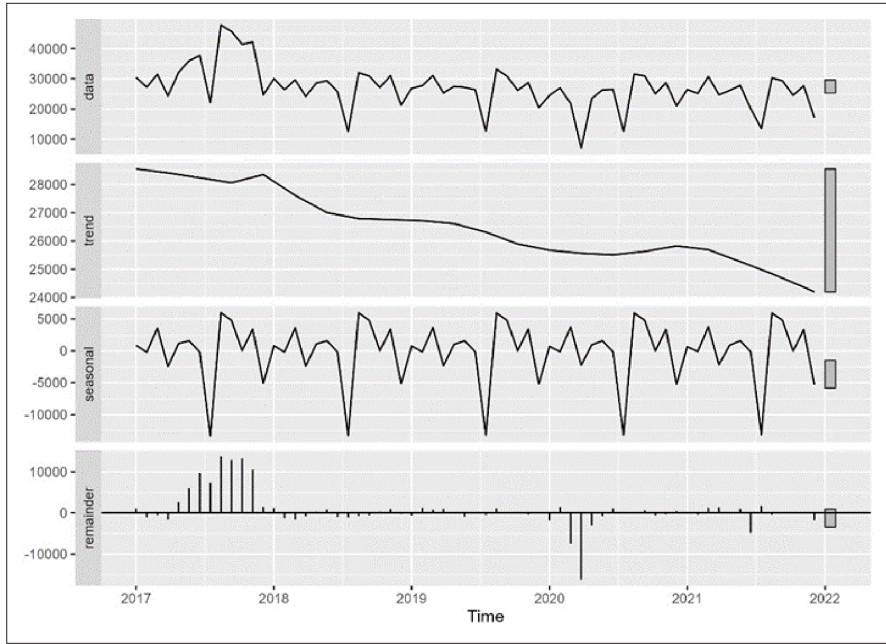

**Figure 5.** Decomposition of time series in number of cervical cancer screens into seasonal, trend, and irregular component, Denmark 2017–2021. Source: Own calculations based on numbers provided by The Danish Health Data Authority.

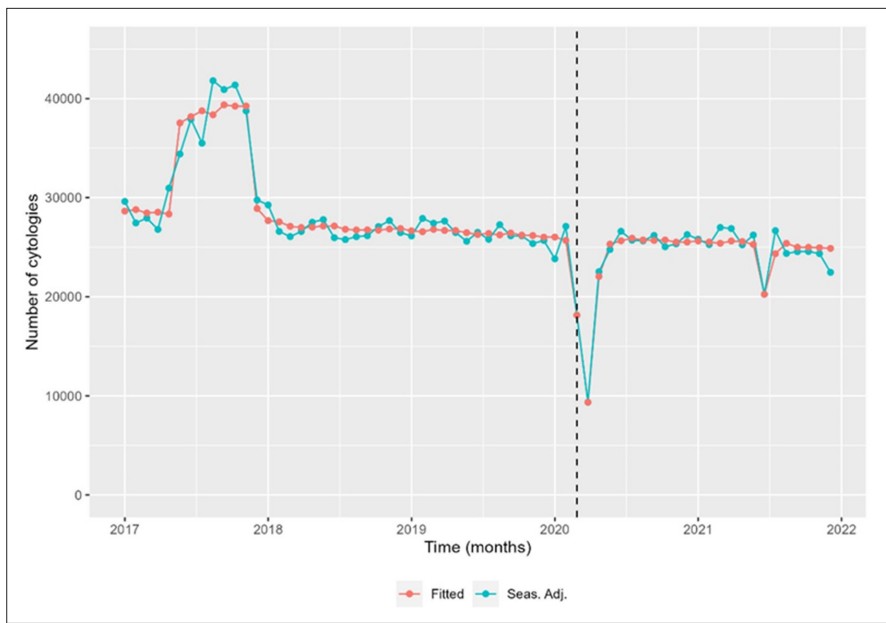

**Figure 6.** Seasonal-adjusted monthly time series and fitted model of the number of cervical cancer screens, Denmark 2017–2021. Source: Own calculations based on numbers provided by The Danish Health Data Authority.

The online version of this article includes the following figure supplement(s) for figure 6:

**Figure supplement 1.** Seasonal-adjusted time series and fitted model of the number of cervical cancer screens, by region 2017–2021.

**Table 3.** Estimated changes in number of seasonally adjusted cervical cell samples. Denmark and regions, 2018–2020. Reductions and increases are with respect to the average base line.

| | Time period | Estimated change in percent |
|---|---|---|
| Denmark | 6 November 2017 – 02 January 2022<br>24 February 2020 – 22 March 2020<br>23 March – 19 April 2020<br>21 June 2021 – 18 July 2021 | –1.9% [CI –2.9 to –0.8] annual reduction<br>–30.7% [CI –36.5 to –24.3]<br>–61.9% [CI –65.2 to –58.2]<br>–20.3% [CI –27.1 to –12.9] |
| Capital | 24 February 2020 – 22 March 2020<br>23 March 2020 – 19 April 2020<br>21 June 2021 – 18 July 2021 | –32.4% [CI –38.1 to –26.1]<br>–57.9% [CI –61.7 to –53.7]<br>–23.8% [CI –30.3 to –16.8] |
| Central | 24 February 2020 – 22 March 2020<br>23 March 2020 – 19 April 2020<br>21 June 2021 – 18 July 2021 | –31.0% [CI –37.6 to –23.8]<br>–62.8% [CI –66.6 to –58.5]<br>–24.0 [CI –31.2 to –16.0] |
| North | 09 September 2019 – 02 January 2022<br>24 February 2020 – 22 March 2020<br>23 March 2020 – 19 April 2020<br>29 March 2021 –25 April 2021 | –6.9% [CI –8.9 to –4.8] annual reduction<br>–33.9% [CI –41.5 to –25.3]<br>–61.1% [CI –65.9 to –55.5]<br>94.5% [CI 72.9–118.8] |
| South | 01 January 2018 – 02 January 2022<br>23 March 2020 – 19 April 2020<br>21 June 2021 – 18 July 2021 | –5.0% [CI –6.4 to –3.6] annual reduction<br>–60.4% [CI –64.4 to –55.9]<br>–19.6 [CI –27.6 to –10.8] |
| Zealand | 23 March 2020 – 19 April 2020<br>01 March 2021 – 28 March 2021<br>21 June 2021 – 18 July 2021 | –67.6% [CI –71.8 to –62.7]<br>31.9% [CI 16.0–50.0]<br>–12.7% [CI –23.4 to –0.6] |

Source: Own calculations based on numbers provided by The Danish Health Data Authority.

which the model illustrated in *Figures 4–6* and *Table 1*. There was a drop in activity of –30.7% [CI –36.5 to –24.3] from 24 February 2020 to 22 March 2020 of –61.9% [CI –65.2 to –58.2] from 23 March 2020 to 19 April 2020 (*Table 3*). Hereafter, activity returned to the previously (linearly decreasing) level. Unexpectedly, there was a drop in activity from 21 June 2021 to 18 July 2021 of –20.3% [CI –27.1 to –12.9] (*Figure 5*, *Table 3*).

When the data from Denmark were split by region, the numbers from Capital Region, Region Zealand, and Region Central followed the national trend (*Figure 6—figure supplement 1*, *Table 3*). Region North had an annual reduction of –6.9% [CI –8.9 to –4.8] from 9 September 2019 to 2 January 2022, and Region South of –5.0% [CI –6.4 to –3.6] from 1 January 2018 to 2 January 2022 (*Figure 6—figure supplement 1*, *Table 3*). The drop around 1 July 2021 was seen in all regions except Region North.

## Discussion
### Main finding
A survey undertaken by the International Cancer Screening Network documented that Denmark was one of the few countries where the health authorities did not suspend cancer screening during the COVID-19 pandemic (*Puricelli Perin et al., 2021*). Here, we demonstrated that this policy was largely followed, as only a brief drop was seen in the screening activity following the lockdown of the society on 13 March 2020.

Our analysis also illustrated the difficulties of using time series analysis in assessing a causal association between the COVID-19 pandemic and screening activity, as activity patterns were also affected by other societal circumstances, such as the shortage of radiologists.

For cervical screening, except during the unique initiative for screening of elderly women in 2017, our data indicated a slight, statistically significant downward trend in cervical screening activity. During the last 10 years, Denmark has experienced a slight general decline in participation in cervical screening. The health authorities are aware of this tendency, and self-sampling kits for HPV testing are currently being rolled out as an alternative to clinician collected samples (*DKLS, 2022*). A more detailed analysis showed that this was limited to two of the five Danish regions. Further in-depth analyses are required to explain this phenomenon. Furthermore, in mid-2021, there was a temporal drop in activities in four out of five regions. It coincided with a national nurse strike, which is unlikely to be directly linked to the drop, as the strike did not involve nurses employed in GP/gynaecologist practices. It could have affected the minor part of the screening activity that the hospitals undertake.

For mammography screening, our data showed activity decline in relation to the first lockdown and that the pre-lockdown levels were reached after several month. This could be due to the issue of extra reminders for screening, but further research would be needed to analyse the association, for example, through a survey. Further, we saw a steep decrease in the screening activity during the second half of 2021, limited to two of the five Danish regions.

In Denmark, the same staff works in screening and diagnostic mammography, and the Capital Region decided to limit the number of screening invitations in August–October 2021 to 25% of the normal level to meet the time limits for diagnostic mammography set in the Danish breast cancer patient package. Incidentally, this was well captured by our model. During this period, the mammography service in the Capital Region underwent organisational changes, moved location, acquired new equipment, and implemented an AI reader.

In our analysis, we focused on the potential effect of the COVID-19 epidemic on breast and cervical screening activity. One can speculate that the pandemic might have affected waiting times for follow-up and treatment of detected lesions. To answer this question, further in-depth analysis of health care data would be needed. However, it should be stressed that reminder systems for follow-up of abnormal findings were in place also during the pandemic.

### Strength and limitations
The main strength of this study was the usage of nationwide register data.

In Denmark, all health care activities are registered in national databases. During the COVID-19 pandemic, the Danish Health Authority used the Danish Health Insurance Register to monitor cervical screening. In Denmark, GPs and office-based gynaecologists are fee-for-service paid by the

government, and these payments are registered in the Danish Health Insurance Register. Therefore, the completeness of the data depends on the screening providers' use of correct payment codes and timely reporting. In the annual monitoring of cervical screening, undertaken by the regions and reported in DKLS, data are retrieved from the National Pathology Register, which covers all pathology specimens analysed in Denmark. Nationwide, there was a difference between the two datasets in 2021, with 323,598 cell samples in our dataset and 372,508 in DKLS, (*Supplementary files 2*). As the DKLS data are expected to be complete, 13% of cell samples from 2021 were thus missing in our data, but the time trends are expected to be the same.

The Danish Health Authority used the National Patient Register as a source for the COVID-19 monitoring reports. DKMS use the same data source. As our data are reported by year, and DKMS' data are reported by screening invitation-round, the numbers are not directly comparable. However, when corrected for length of period, we had 5% more screens than DKMS, (*Supplementary files 2*). The deficit in DKMS data derived almost entirely from two of the five Danish regions, which might partly be due to different dates for data retrieval.

A further limitation was that we used the number of tests to analyse trends, not considering that the size of the screening target groups could vary slightly over the years. It should also be noted that it is impossible to make a causal conclusion based on a trend analysis.

Another limitation was that we reported on tests performed and not on women examined.

Finally, even though a number of tests have been carried out to check the validity of our analyses (i.e. analysis of (partial) autocorrelation functions, periodogram of Pearson residuals, non-randomised PIT histogram, marginal calibration plot, normality of residuals), our results should be further checked in different settings, such as different time series models and/or using data with higher time resolution (weekly, daily).

## Other studies

During the first 3 months of the COVID-19 pandemic in Denmark, the number of cancer diagnoses dropped by one-third compared with previous years (*Skovlund et al., 2020*). Early during the lockdown, the Danish Cancer Society collected information from screening providers and reported decreased participation in mammography and cervical screening (*Danish Cancer Society, 2020a*; *Danish Cancer Society, 2020b*). According to the DKLS 2020 report, there was a drop in cervical screens by approximately 10% from 2019–2020 is suggested to be related to COVID-19 and the lockdown (*Waldstrøm et al., 2022*).

In the Netherlands, suspension of the breast cancer screening programme was associated with a substantial decrease in women diagnosed with breast cancer in the first weeks after suspension, and the number remained low until June 2020 (*Dinmohamed et al., 2020*). Australia experienced decreased capacity after resuming the breast screening program, only reaching 83% of the 2018 level (*Feletto et al., 2020*). This could eventually delay screening and result in disease progression (*Davies et al., 2022*; *Wilson et al., 2021*).

In the UK, Wilson et al. undertook a mixed-method study in August–November 2020, comprising an online survey and qualitative interviews. In the survey, 30% answered that they were less likely to attend cervical screening than before the corona pandemic lockdown, and 75% declared that they were worried about delay in cancer screening caused by COVID-19 (*Wilson et al., 2021*). Although reluctance to participate in cervical screening is not a new phenomenon, especially among young women and college graduates (*Wenger et al., 2022*), further cancellation related to the pandemic could potentially have long-term consequences, as participation in screening once is a predictor of participation in the future (*Kotzur et al., 2020*; *Wilson et al., 2021*). The pandemic could therefore in particular be expected to affect women scheduled to enter the screening program during the pandemic (*Wilson et al., 2021*). Moreover, vulnerable groups who have been advised to be careful during the pandemic would weigh the pros and cons of screening against the risk of COVID-19 infection (*Walker et al., 2021*).

## Public health implications

Our analysis showed a steep, temporal drop in breast and cervical screening activity following the WHO declaration of the pandemic and the simultaneous lockdown of the Danish society, indicating that COVID-19 did have an imitated and temporary effect on breast and cervical cancer screening in

Denmark. The focused policy of keeping screening going was a decisive factor. Political decisions were communicated from the Danish Health Authority to the five regions responsible for cancer screening. The Danish Cancer Society and the press also played an essential role in repeatedly reporting cancer screening status. The initial suspension in marts 2020 of cervical screening undertaken by GPs was revoked within 9 days, and extra reminders for mammography screening were issued when real-time monitoring data indicated a decrease in activity.

The very possibility of continuing screening during the pandemic was undoubtedly facilitated by the robustness of the health care system. During the period, March 2020–December 2021, 3550 persons in Denmark died with COVID-19 (*Neergaard, 2022*), constituting 3.6% out of a total number of 99,000 deaths in the period (*StatBank Denmark, 2022*). Life expectancy increased from 2018/2019 to 2019/2020 for both men and women but levelled off from 2019/2020 to 2020/2021; for men changing from 79.5 to 79.6 years, and for women slightly decreased from 83.6 to 83.4 years (*StatBank Denmark, 2022*). The COVID-19 lockdown period was nevertheless a period where health care resources were stretched far beyond the normal level. After a breakdown of negotiations on a new agreement, the nurses went on strike in June–July 2021, causing further delays in elective treatment, for example, hip replacement (*TV2 News, 2022*).

As handling cancer screening during an extraordinary situation like the COVID-19 pandemic depends on the local organisation and resources, the experiences from Denmark can probably not be generalised to all settings. Nevertheless, the example illustrated the potentials for health policy implementation in a high-income, welfare state with a publicly run health care system.

## Conclusion

In Denmark, it was decided politically to continue mammography and cervical screening throughout the COVID-19 pandemic in 2020 and 2021. Our study showed that screening activity dropped suddenly at the time of the first lockdown of the society but recovered in the following months and went back to the pre-pandemic level.

## Acknowledgements

The Danish Health Data Authority provided data for the study. Region Zealand financially supported the study (grant number: R22-A597). The funder had no impact on the study design and interpretation of data.

## Additional information

### Competing interests

Mette Hartmann Nonboe: received HPV-test-kits free of charge for a method study from Roche. The author has no other competing interests to declare. Elsebeth Lynge: received HPV-test kits from Roche for a method study. The remaining authors have no conflicts of interest to declare. The other authors declare that no competing interests exist.

### Funding

| Funder | Grant reference number | Author |
| --- | --- | --- |
| Region Zealand | R22-A597 | Elsebeth Lynge |

The funders had no role in study design, data collection and interpretation, or the decision to submit the work for publication.

### Author contributions

Mette Hartmann Nonboe, Data curation, Investigation, Writing – original draft, Project administration; George Napolitano, Software, Formal analysis, Investigation, Visualization, Methodology, Writing – review and editing; Jeppe Bennekou Schroll, Ilse Vejborg, Marianne Waldstrøm, Validation, Writing – review and editing; Elsebeth Lynge, Conceptualization, Resources, Supervision, Funding acquisition, Writing – original draft, Writing – review and editing

Author ORCIDs
Mette Hartmann Nonboe http://orcid.org/0000-0003-0085-9109

## Decision letter and Author response
Decision letter https://doi.org/10.7554/eLife.81605.sa1
Author response https://doi.org/10.7554/eLife.81605.sa2

## Additional files

### Supplementary files
• MDAR checklist
• Supplementary file 1. Overview of dates within "Months".
• Supplementary file 2. Comparison of study data with DKMS and DKLS Reports 2021.

### Data availability
The Danish Health Data Authority has provided all data. All data generated or analysed during this study are included in the manuscript and supporting file. Dataset has been deposited at DataCite (https://search.datacite.org/) and with DOI:10.17894/ucph.b1860a53-63e3-45c7-970b-6482ab2947c7. All analyses and plots were done using R ver. 4.1.1 (R Core Team; 2021), with the package collection tidyverse (Wickham et al.; 2019) and the packages tscount (Liboschik et al.; 2017) and tsoutlier (ópez-de-Lacalle; 2019). The software and packages are publicly available and can be downloaded at https://cran.r-project.org/mirrors.html, https://cran.r-project.org/web/packages/tsoutliers/index.html and https://www.tidyverse.org/.

The following dataset was generated:

| Author(s) | Year | Dataset title | Dataset URL | Database and Identifier |
|---|---|---|---|---|
| Nonboe MH | 2022 | Data | https://doi.org/10.17894/ucph.b1860a53-63e3-45c7-970b-6482ab2947c7 | Electronic Research Data Archive University of Copenhagen, 10.17894/ucph.b1860a53-63e3-45c7-970b-6482ab2947c7 |

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
