## [Editor Report]

Denmark was one of the few countries that did not suspend cancer screening in the early stage of the COVID-19 pandemic. This important study offers convincing evidence of how the pandemic impacted the use of breast and cervical cancer screening services. This article has a broad interest to public health researchers and health policy implementation.

---

## [Decision Letter]

**Decision letter after peer review:**

Thank you for submitting your article "Impact of COVID-19 pandemic on cancer screening in Denmark: A register-based study" for consideration by *eLife*. Your article has been reviewed by 3 peer reviewers, and the evaluation has been overseen by a Reviewing Editor and Eduardo Franco as the Senior Editor. The following individuals involved in the review of your submission have agreed to reveal their identity: Douglas M Puricelli Perin (Reviewer #1); Johannes Berkhof (Reviewer #2).

As is customary in *eLife*, the reviewers have discussed their critiques with one another and with the editors. What follows below is the Reviewing Editor's edited compilation of the essential and ancillary points provided by reviewers in their critiques and in their interaction post-review. Please submit a revised version that addresses these concerns directly. Although we expect that you will address these comments in your response letter, we also need to see the corresponding revision clearly marked in the text of the manuscript. Some of the reviewers' comments may seem to be simple queries or challenges that do not prompt revisions to the text. Please keep in mind, however, that readers may have the same perspective as the reviewers. Therefore, it is essential that you attempt to amend or expand the text to clarify the narrative accordingly.

Essential revisions:

1) The reviewers all concur this is a strong article that merits publication with suggested edits to improve clarity and presentation, which should be addressed individually.

2) Appropriateness of the time series model should also be discussed, including alternative approaches that might improve fit, such as using exact dates in line with important events of the pandemic.

*Reviewer #1 (Recommendations for the authors):*

In this paper, the authors assessed through a time series analysis of the total number of breast and cervical cancer screening tests, stratified by region, in Denmark. The completeness and reliability of the registry databases and adequate modeling tools are strengths of this study, while the reliance on a number of tests as the outcome measure without information about coverage and target populations presents as a weakness.

The authors adequately described the methods used in their research, which would allow researchers to conduct similar and more in-depth studies following these analyses utilizing these databases. The results were presented in a thorough manner, highlighting the immediate impact that the declaration of the COVID-19 pandemic had on people seeking health services, considering that mammography abruptly dropped by 43.3% and cervical cell samples 61.9% in the initial month of the pandemic.

Although I understand the authors' attempt to share the good news about the number of tests coming up after this initial shock, I would argue that they have not quickly recovered to pre-pandemic levels across all regions, especially considering the numbers from 2017 and 2018, a steep decrease in mammography in the second half of 2021, and a downward trend in cervical cancer screening, even if not fully explained by the pandemic.

This work adds to the general literature on cancer screening in the COVID-19 era, and even though we cannot directly infer it from the analysis, this work provides great insight into how a disruptive event such as a pandemic may prevent people from seeking health services during and long after the event, considering that Denmark's services did not stop and the health resources and infrastructure were readily available for cancer screening coupled with strong messaging from government and advocacy groups encouraging people to screen.

Thank you for the opportunity to review your paper as it contributes greatly to understanding the effects of the ongoing COVID-19 pandemic on cancer screening in a country that did not have its screening services interrupted. It is very illuminating, but not unexpected, to see the reduction in the number of breast and cervical cancer screening tests even if services were not interrupted and strongly promoted. I have a few suggestions that hopefully would help strengthen your article:

1. I suggest redefining your study purpose as described on page 3, lines 81-82: "In the present study, we assessed the population's cancer screening uptake during the COVID-19 pandemic in Denmark." It is my understanding that you did not assess patient uptake but looked at the number of tests instead, even though this could be understood as a proxy, so it would more appropriately describe what you did. I would also advise framing it around the research question that your paper is seeking to answer, which I understand as "have we seen effects of the COVID-19 pandemic on mammography and cervical screening considering the number of tests performed nationally and by region?" It would be best to specifically nominate breast and cervical cancer screening since you did not analyze bowel screening.

2. Your methods are thoroughly described and it was particularly helpful to have a brief description of the Danish databases and screening programs. I would like to focus on the bowel cancer screening programs, which you also describe in the methods even though you did not include it in the analysis. On page 4, lines 106-109, you state: "As only activities in the hospitals and the private sector were monitored, activities in the bowel cancer screening program were not included, as this program is based on self-sampled tests." I understand your explanation about why the bowel program was not included but I find that it would be more adequate to say that they were not included because the FIT kits are sent to the person's home instead, considering that self-sample tests could be performed in clinics. In addition, I still wonder about follow-up colonoscopies to positive FIT tests, as I would consider that part of the screening process and a second step in the bowel screening event. Would an analysis of follow-up colonoscopies have been feasible within the Danish databases? If so, I would suggest including them in this paper. Otherwise, I agree on keeping the focus on breast and cervical cancer screening, making that clear in the title and introduction, and would remove the description about the bowel program.

3. In the results, I would suggest strictly reporting the outcomes of the analyses and avoiding making explanatory comments, as you could better expand on those in the discussion. For example, on page 9, lines 199-202, you state: "The activity decline was short-termed; however, it did not reach the previous levels in the first months after the first lockdown. Pre-lockdown levels were reached only after the issue of additional extra reminders for examination." The number of mammographies did recover but it is not obvious to me looking at your data that the decline is short-termed, considering that there was another big drop in 2021 and that the numbers seem to still be going down (although not statistically significant). Also, it is not immediately obvious where the extra reminders fit in your results since you cannot directly attribute the catch-up in the number of tests to the reminders based on the time series analysis only. It would be appropriate to suggest an effect in the discussion, however, as you did, and maybe encourage researchers to look further into that through a survey, for example.

4. Finally, the discussion would benefit from more insights and details regarding the regional differences in the number of screening tests and the downward trend in cervical cancer screening.

On page 19, lines 295-300, you note that "A more detailed analysis showed that this was limited to two of the five Danish regions. We do not know the reason for this phenomenon. Furthermore, there were temporal drop-in activities in four out of five regions. It coincided with a national nurse strike, which is unlikely to be directly linked to the drop, as the strike did not involve nurses employed in GP/gynaecologist practices. It could have affected the minor part of the screening activity that the hospitals undertake." That definitely encourages further research but would you have any suggestions as to why? Could it be exacerbated by the lingering pandemic? Is this trend also seen for other health services within the same area?

Also, the large increase in number of mammography in Region South observed in 2020 and 2021 shown in Table 1 is not explained in the discussion. Was there a migration within Denmark? People moving to other regions considering the pandemic, wanting to be closer to family, now being able to work from home? These are questions that come to my mind as I look at your results.

One final comment, I would avoid a definite statement about COVID-19 not affecting cancer screening in Denmark. I would rephrase the statement between pages 22-23, lines 369-371 to say that the pandemic did have an effect even if temporarily, and I would limit that observation to breast and cervical cancer screening since you did not include an analysis of bowel cancer screening. And in your conclusion, on page 23, lines 396-398, I would not agree that your study shows that screening activity rapidly went back to pre-pandemic level, but that it did recover after a few months from a sudden and steep drop when the pandemic was declared.

*Reviewer #2 (Recommendations for the authors):*

This paper estimates the effect of the Covid19 lockdown on the number of screens in the breast cancer and cervical cancer screening program in Denmark, where the program was not stopped after the start of the pandemic. The authors give several reasons for a decline in the number of screens in the discussion of their paper, including concern about becoming infected, concern about unnecessary burdening on the health system, etc.

The main strengths of the paper are the high quality of the registry data and the advanced statistical approach for estimating the Covid19-related drop in the screening coverage. A statistical time series model was applied to estimate the change in coverage after the start of the pandemic, adjusted for seasonal variation. The statistical analyses are advanced, but the appropriateness of the statistical model has not been checked. Nevertheless, the model seems to provide strong evidence of a short, steep decline in coverage after the start of the Covid19 pandemic. The authors are very cautious in not overinterpreting the data, for instance, by repeatedly stating that it is difficult to draw causal conclusions based on time series data. A causal interpretation of the results can be strengthened by better connecting the breast cancer and cervical cancer screening coverage data. The two cancers were analyzed and discussed independently but a causal association seems plausible since for both cancers a sharp temporary decline in the number of screens was observed after the start of the pandemic. Moreover, separate time series models fitted to the different regions in Denmark showed a similar decline in coverage right after the start of the pandemic, which again strengthens the interpretation of the presented results. A limitation of the paper is that it does not provide information on the effect of Covid19 on follow-up procedures for screen-positive women.

The data are interesting and the statistical analyses are advanced. The following, mainly methodological comments can be raised.

1. The appropriateness of the GARCH model is not discussed. Why has the GARCH model been chosen and it is appropriate when adjusting the data for seasonal variation? The model does not do a very good job of fitting breast cancer screening data in region North (Figure S1): The decline in the time series data after the start of the pandemic was missed by the model. Therefore, it would be interesting to compare the fit of the GARCH model to other time series models.

2. How are the estimated reductions in the number of screens and their 95% confidence intervals measured?

3. The y-axes of some figures do not include the value 0. This gives an exaggerated picture of the reduction in the number of screens and makes it difficult to compare different figures.

4. Figure 2 is a nice decomposition of the breast cancer screening data. A similar figure might be included for cervical cancer.

5. The interpretation of the cervical cancer plot is incomplete. The strong increase in screens in 2017 is not mentioned in the Results section. Besides, the interpretation of the region-specific cervical cancer plots (S3) is incomplete. The reduction in all regions except region North in July 2021 is mentioned, but the large increase in region North in April 2021 is not mentioned.

6. The discussion about registries in Table 4 seems a bit technical. Maybe it should be moved to the Supplementary Appendix.

7. Did the Covid19 influence follow-up procedures in Denmark? If the number of screens remained the same but follow-up medical procedures (diagnostics + treatment) were postponed, will this not lead to long waiting times that could have been avoided by adjusting the number of screens downwards?

*Reviewer #3 (Recommendations for the authors):*

In this study, the authors aim to show how mammography screening and cervical cancer screening in Denmark were affected by the covid-19 pandemic. By using data from national registries and analyzing the data using a time-series approach, they find that a drop in attendance during the first lock-down period, although screening was not suspended. They also found that the screening attendance rapidly returned to the pre-pandemic level in 2020.

Strengths:

The use of data from national register data, i.e. data that covers the whole Danish population and is not suffering from selection bias is a major strength of this paper. The data analysis, using a seasonally adjusted time series, identifying seasonal trends and patterns are sound and appropriate methodology for this type of data. The authors have also handled problems with outliers by both analytical and visual inspection.

Weaknesses:

As the authors clearly state, it is not possible to draw causal conclusions from trend analysis. However, with the pandemic having so severe consequences, it is highly likely that the change in the observed data is a consequence of the pandemic. Although the study analyses data from national registries there are some missing data, but not to an extent that would change the conclusions in the paper.

There are two major weaknesses of the paper. Firstly, the trend analysis is very data-driven and the actual dates of the first lock-down are not explicitly taken into the model. The first lockdown was effectuated on March 13th, 2020, while the results presented found a decrease of 22.6% from February 24 to March 22nd, 2020, and a decrease of 43.3% from March 23rd to April 19th, 2020 compared to pre-pandemic levels. The 22.6% drop is then estimated for both pre and during the lockdown, and one would not expect a decrease in February due to the lockdown in March. A more "mix" approach forcing the dates of the actual lock-down in the data analysis would probably give a better estimate of the effect. The second weakness is that subsequent covid measures during 2020 and 2021 such as enforcing masks/face shields in the autumn of 2020, and the subsequent lockdown at the end of 2021 are not mentioned in the paper.

The abstracts lack information on the data analysis and the conclusion should rather focus on empirical findings rather than the Danish policy of continuing screening.

The authors should consider using the exact date of the lockdown in the data analysis rather than being fully data-driven, or at least trying to explain a reduction in screening activity before the lockdown. The comparisons in yearly screening, i.e. comparing 2019 with 2020, should rather be broken down more in line with important dates of the epidemic.

---

## [Author Response]

Essential revisions:1) The reviewers all concur this is a strong article that merits publication with suggested edits to improve clarity and presentation, which should be addressed individually.

Thank you for the positive comments.

2) Appropriateness of the time series model should also be discussed, including alternative approaches that might improve fit, such as using exact dates in line with important events of the pandemic.

For discussion on the statistical method, see Answer 1 under Reviewer 2 below.

Reviewer #1 (Recommendations for the authors):In this paper, the authors assessed through a time series analysis of the total number of breast and cervical cancer screening tests, stratified by region, in Denmark. The completeness and reliability of the registry databases and adequate modeling tools are strengths of this study, while the reliance on a number of tests as the outcome measure without information about coverage and target populations presents as a weakness.The authors adequately described the methods used in their research, which would allow researchers to conduct similar and more in-depth studies following these analyses utilizing these databases. The results were presented in a thorough manner, highlighting the immediate impact that the declaration of the COVID-19 pandemic had on people seeking health services, considering that mammography abruptly dropped by 43.3% and cervical cell samples 61.9% in the initial month of the pandemic.Although I understand the authors' attempt to share the good news about the number of tests coming up after this initial shock, I would argue that they have not quickly recovered to pre-pandemic levels across all regions, especially considering the numbers from 2017 and 2018, a steep decrease in mammography in the second half of 2021, and a downward trend in cervical cancer screening, even if not fully explained by the pandemic.This work adds to the general literature on cancer screening in the COVID-19 era, and even though we cannot directly infer it from the analysis, this work provides great insight into how a disruptive event such as a pandemic may prevent people from seeking health services during and long after the event, considering that Denmark's services did not stop and the health resources and infrastructure were readily available for cancer screening coupled with strong messaging from government and advocacy groups encouraging people to screen.

Thank you for these reflections. Please see our responses to the points below.

Thank you for the opportunity to review your paper as it contributes greatly to understanding the effects of the ongoing COVID-19 pandemic on cancer screening in a country that did not have its screening services interrupted. It is very illuminating, but not unexpected, to see the reduction in the number of breast and cervical cancer screening tests even if services were not interrupted and strongly promoted. I have a few suggestions that hopefully would help strengthen your article:

Thank you for your positive and very helpful comments.

1. I suggest redefining your study purpose as described on page 3, lines 81-82: "In the present study, we assessed the population's cancer screening uptake during the COVID-19 pandemic in Denmark." It is my understanding that you did not assess patient uptake but looked at the number of tests instead, even though this could be understood as a proxy, so it would more appropriately describe what you did. I would also advise framing it around the research question that your paper is seeking to answer, which I understand as "have we seen effects of the COVID-19 pandemic on mammography and cervical screening considering the number of tests performed nationally and by region?" It would be best to specifically nominate breast and cervical cancer screening since you did not analyze bowel screening.

Answer 1. We agree to specify the study purpose as follows, on page 3 line 81-83:

“In the present study, we assessed the effect of the COVID-19 pandemic on the number of tests performed in mammography and cervical screening, nationally and by region.”

2. Your methods are thoroughly described and it was particularly helpful to have a brief description of the Danish databases and screening programs. I would like to focus on the bowel cancer screening programs, which you also describe in the methods even though you did not include it in the analysis. On page 4, lines 106-109, you state: "As only activities in the hospitals and the private sector were monitored, activities in the bowel cancer screening program were not included, as this program is based on self-sampled tests." I understand your explanation about why the bowel program was not included but I find that it would be more adequate to say that they were not included because the FIT kits are sent to the person's home instead, considering that self-sample tests could be performed in clinics. In addition, I still wonder about follow-up colonoscopies to positive FIT tests, as I would consider that part of the screening process and a second step in the bowel screening event. Would an analysis of follow-up colonoscopies have been feasible within the Danish databases? If so, I would suggest including them in this paper. Otherwise, I agree on keeping the focus on breast and cervical cancer screening, making that clear in the title and introduction, and would remove the description about the bowel program.

Thank you for your suggestion. Sadly, the follow-up colonoscopy data were not part of the Danish health care monitoring system from which we extracted data. Therefore, we have followed your suggestion and removed the description of the bowel program. And added the text:

Setting, page 4, line 98-101:

“Lastly, biennially screening for bowel cancer. As the this program is based on faecal immunological test kits sent to people’s home, the program was not part of the Danish health care monitoring system from which we extracted data.”

3. In the results, I would suggest strictly reporting the outcomes of the analyses and avoiding making explanatory comments, as you could better expand on those in the discussion. For example, on page 9, lines 199-202, you state: "The activity decline was short-termed; however, it did not reach the previous levels in the first months after the first lockdown. Pre-lockdown levels were reached only after the issue of additional extra reminders for examination." The number of mammographies did recover but it is not obvious to me looking at your data that the decline is short-termed, considering that there was another big drop in 2021 and that the numbers seem to still be going down (although not statistically significant). Also, it is not immediately obvious where the extra reminders fit in your results since you cannot directly attribute the catch-up in the number of tests to the reminders based on the time series analysis only. It would be appropriate to suggest an effect in the discussion, however, as you did, and maybe encourage researchers to look further into that through a survey, for example.

Answer 3. We have removed the sentence as suggested from the results on page 9, and added the following text in the discussion on page 19, line 294-298:

“For mammography screening, our data showed activity decline in relation to the first lockdown and that the pre-lockdown levels were reached after several month. This could be due to the issue of extra reminders for screening, but further research would be needed to analyse the association, e.g. through a survey. Further, we saw a steep decrease in the screening activity during the second half of 2021, limited to two of the five Danish regions.”

4. Finally, the discussion would benefit from more insights and details regarding the regional differences in the number of screening tests and the downward trend in cervical cancer screening.

During the last ten years, Denmark has experienced a slight general decline in participation in cervical screening, which is also reported by the Danish Quality Database for Cervical Cancer Screening.

We have added the following text to the Discussion section:

Main finding, page 19, line 283-287:

“During the last ten years, Denmark has experienced a slight general decline in participation in cervical screening. The health authorities are aware of this tendency, and self-sampling kits for HPV-testing are currently being rolled out as an alternative to clinician collected samples (*Danish Quality Database for Cervical Cancer Screening. DKLS Report 2021*; *2022*).”

Line 288-289:

“Further in-depth analyses are required to explain this phenomenon.”

On page 19, lines 295-300, you note that "A more detailed analysis showed that this was limited to two of the five Danish regions. We do not know the reason for this phenomenon. Furthermore, there were temporal drop-in activities in four out of five regions. It coincided with a national nurse strike, which is unlikely to be directly linked to the drop, as the strike did not involve nurses employed in GP/gynaecologist practices. It could have affected the minor part of the screening activity that the hospitals undertake." That definitely encourages further research but would you have any suggestions as to why? Could it be exacerbated by the lingering pandemic? Is this trend also seen for other health services within the same area?

After the nurse strike, there have been a notable increase in nurses leaving the Danish Health Care system, with the theoretical consequence of affecting several other activities. And this could potentially be exacerbated by the lingering pandemic, as the toll on the remaining nurses is substantial. However, it has not been possible to find any research regarding this for the time being.

Also, the large increase in number of mammography in Region South observed in 2020 and 2021 shown in Table 1 is not explained in the discussion. Was there a migration within Denmark? People moving to other regions considering the pandemic, wanting to be closer to family, now being able to work from home? These are questions that come to my mind as I look at your results.

To the best of our knowledge, there has not been a notable migration within Denmark under COVID-19. However, your comments made us aware of a mistake, we had made in Table 1. We had unfortunately reported the numbers for Region South for year 2017 to 2019 in the row for Region Zealand and vice versa. This has now been corrected and we deeply apologise for this mistake. We have afterwards looked through all tables and checked the numbers again to make sure that a similar mistake was not made elsewhere, and some minor mistakes in numbers have been adjusted.

There is therefore no longer a large increase in the number of mammographies in Region South.

One final comment, I would avoid a definite statement about COVID-19 not affecting cancer screening in Denmark. I would rephrase the statement between pages 22-23, lines 369-371 to say that the pandemic did have an effect even if temporarily, and I would limit that observation to breast and cervical cancer screening since you did not include an analysis of bowel cancer screening. And in your conclusion, on page 23, lines 396-398, I would not agree that your study shows that screening activity rapidly went back to pre-pandemic level, but that it did recover after a few months from a sudden and steep drop when the pandemic was declared.

We have thought-out the manuscript changed the focus to breast and cervical screening, as suggested. Moreover, taking your comments on the effect on screening in Denmark into account, we have changed the wording on pages 23-24, line 378-381. We suggest the following changes:

“Our analysis showed a steep, temporal drop in breast and cervical screening activity following the WHO declaration of the pandemic and the simultaneous lockdown of the Danish society, indicating that COVID-19 had a limited and temporary effect on breast and cervical screening in Denmark.”

We have changed the last part of the conclusion on page 24, line 406-408 to the following:

“Our study showed that screening activity dropped suddenly at the time of the first lockdown of the society but recovered in the following months and went back to the pre-pandemic level.”

According to your suggestion, we made the following chances to the conclusion in the abstract to fit with the conclusion in the manuscript:

“Denmark continued screening during the pandemic, but following the first lockdown a temporary drop was seen in breast and cervical screening activity.”

Reviewer #2 (Recommendations for the authors):This paper estimates the effect of the Covid19 lockdown on the number of screens in the breast cancer and cervical cancer screening program in Denmark, where the program was not stopped after the start of the pandemic. The authors give several reasons for a decline in the number of screens in the discussion of their paper, including concern about becoming infected, concern about unnecessary burdening on the health system, etc.The main strengths of the paper are the high quality of the registry data and the advanced statistical approach for estimating the Covid19-related drop in the screening coverage. A statistical time series model was applied to estimate the change in coverage after the start of the pandemic, adjusted for seasonal variation. The statistical analyses are advanced, but the appropriateness of the statistical model has not been checked. Nevertheless, the model seems to provide strong evidence of a short, steep decline in coverage after the start of the Covid19 pandemic. The authors are very cautious in not overinterpreting the data, for instance, by repeatedly stating that it is difficult to draw causal conclusions based on time series data. A causal interpretation of the results can be strengthened by better connecting the breast cancer and cervical cancer screening coverage data. The two cancers were analyzed and discussed independently but a causal association seems plausible since for both cancers a sharp temporary decline in the number of screens was observed after the start of the pandemic. Moreover, separate time series models fitted to the different regions in Denmark showed a similar decline in coverage right after the start of the pandemic, which again strengthens the interpretation of the presented results. A limitation of the paper is that it does not provide information on the effect of Covid19 on follow-up procedures for screen-positive women.

Thank you for these reflections. Please see our responses to the points below.

The data are interesting and the statistical analyses are advanced. The following, mainly methodological comments can be raised.

Thank you for your positive comments.

1. The appropriateness of the GARCH model is not discussed. Why has the GARCH model been chosen and it is appropriate when adjusting the data for seasonal variation? The model does not do a very good job of fitting breast cancer screening data in region North (Figure S1): The decline in the time series data after the start of the pandemic was missed by the model. Therefore, it would be interesting to compare the fit of the GARCH model to other time series models.

In order to check the validity of the model, a number of tests were performed:

– Before the model fitting, the autocorrelations and partial autocorrelations of the seasonal-adjusted time series were analysed, in order to select the most appropriate model (i.e. previous observations and conditional means to be regressed on).

– The fitted model was evaluated through cumulative periodogram of Pearson residuals, non-randomized PIT histogram and marginal calibration plot. These tests were all considered passed by visual inspection.

– The model’s residuals were checked for normality through normal QQ plot and Shapiro-Wilk test. By both the above tests, all residuals appeared normally distributed

The above was not reported in the manuscript to reduce technicalities.

Regarding the North Region, in our interpretation the failure to detect the decline after the start of the pandemic is mostly due to the wide oscillations in the time series before the pandemic, which somehow prevented the tests for outliers to detect the drop in the early pandemic period.

Nonetheless, we agree with the Reviewer that it would be interesting to compare the GARCH model with other time series models, which in principle might fit the data better. However, we believe that such comparison would go far beyond the aims of the present study. We added to the limitations of the study the following, page 21 line 337-341:

“Finally, even though a number of tests have been carried out to check the validity of our analyses (i.e. analysis of (partial) autocorrelation functions, periodogram of Pearson residuals, non-randomized PIT histogram, marginal calibration plot, normality of residuals), our results should be further checked in different settings, such as different time series models and/or using data with higher time resolution (weekly, daily).”

2. How are the estimated reductions in the number of screens and their 95% confidence intervals measured?

Outliers in the seasonal-adjusted time series, once detected, are manually inserted into the regression model as covariates. Estimated reductions are given by 1-exp(β_i) (as the models have been chosen with a logarithmic link function), where β_i is the coefficient of the i-th covariate. The 95% confidence intervals are then obtained from the β’s standard errors using normal approximation. The following has been added to the text on page 8, line 180-182:

“Estimated reductions in the number of screens were obtained from the coefficients of the relevant “intervention” covariate of the model. The corresponding 95% confidence intervals are calculated from the coefficients’ standard error using normal approximation.”

3. The y-axes of some figures do not include the value 0. This gives an exaggerated picture of the reduction in the number of screens and makes it difficult to compare different figures.

Figures have been changed.

4. Figure 2 is a nice decomposition of the breast cancer screening data. A similar figure might be included for cervical cancer.

A new figure was added, kindly see page 16.

We have furthermore deleted the previous Figure 3, as this information was also included in Figure 4, which is now instead Figure 3. Seasonal-adjusted time series and fitted model of the number of mammography screens, Denmark 2017-2021. There is equal numbers of figures related to cervical and mammography screenings, respectively.

5. The interpretation of the cervical cancer plot is incomplete. The strong increase in screens in 2017 is not mentioned in the Results section. Besides, the interpretation of the region-specific cervical cancer plots (S3) is incomplete. The reduction in all regions except region North in July 2021 is mentioned, but the large increase in region North in April 2021 is not mentioned.

The increase in cervical screening activity in 2017 was due to a special initiative for screening of elderly women. The following text has been added on page 14, line 244-245:

“The special initiative for screening elderly women was issued in 2017, which the model illustrates in Figure 4-6 and Table 1”

Even though Region North had a large increase in April 2021, the model detected an annual reduction from 9 September 2019 until 2 January 2022. We do not have an explanation for the large increase in April 2021. However, there is the possibility that it is an artifact in the model.

6. The discussion about registries in Table 4 seems a bit technical. Maybe it should be moved to the Supplementary Appendix.

Table 4 has been moved to Supplementary File.

7. Did the Covid19 influence follow-up procedures in Denmark? If the number of screens remained the same but follow-up medical procedures (diagnostics + treatment) were postponed, will this not lead to long waiting times that could have been avoided by adjusting the number of screens downwards?

We understand the interest in the possible effect of COVID-19 on the follow-up procedures, however the analysis was not possible based on the Danish health care monitoring data. However, we have added the following text, page 20 line 307-313:

“In our analysis, we focused on the potential effect of the COVID-19 epidemic on breast and cervical screening activity. One can speculate that the pandemic might have affected waiting times for follow-up and treatment of detected lesions. To answer this question further in-depth analysis of health care data would be needed. However, it should be stressed that reminder systems for follow-up of abnormal findings were in place also during the pandemic.”

Reviewer #3 (Recommendations for the authors):In this study, the authors aim to show how mammography screening and cervical cancer screening in Denmark were affected by the covid-19 pandemic. By using data from national registries and analyzing the data using a time-series approach, they find that a drop in attendance during the first lock-down period, although screening was not suspended. They also found that the screening attendance rapidly returned to the pre-pandemic level in 2020.Strengths:The use of data from national register data, i.e. data that covers the whole Danish population and is not suffering from selection bias is a major strength of this paper. The data analysis, using a seasonally adjusted time series, identifying seasonal trends and patterns are sound and appropriate methodology for this type of data. The authors have also handled problems with outliers by both analytical and visual inspection.Weaknesses:As the authors clearly state, it is not possible to draw causal conclusions from trend analysis. However, with the pandemic having so severe consequences, it is highly likely that the change in the observed data is a consequence of the pandemic. Although the study analyses data from national registries there are some missing data, but not to an extent that would change the conclusions in the paper.There are two major weaknesses of the paper. Firstly, the trend analysis is very data-driven and the actual dates of the first lock-down are not explicitly taken into the model. The first lockdown was effectuated on March 13th, 2020, while the results presented found a decrease of 22.6% from February 24 to March 22nd, 2020, and a decrease of 43.3% from March 23rd to April 19th, 2020 compared to pre-pandemic levels. The 22.6% drop is then estimated for both pre and during the lockdown, and one would not expect a decrease in February due to the lockdown in March. A more "mix" approach forcing the dates of the actual lock-down in the data analysis would probably give a better estimate of the effect. The second weakness is that subsequent covid measures during 2020 and 2021 such as enforcing masks/face shields in the autumn of 2020, and the subsequent lockdown at the end of 2021 are not mentioned in the paper.

Thank you for these reflections. Please see our responses to the points below.

The abstracts lack information on the data analysis and the conclusion should rather focus on empirical findings rather than the Danish policy of continuing screening.

Thank you for making us aware of this. We adjusted the abstract to better fit the scope of the manuscript as follows:

Method:

A time series analysis was carried out to discover possible trends and outliers in the screening activities in the period 2017-2021.

Conclusions

“Denmark continued cancer screening during the pandemic, but following the first lockdown a temporary drop was seen in breast and cervical screening activity”.

The authors should consider using the exact date of the lockdown in the data analysis rather than being fully data-driven, or at least trying to explain a reduction in screening activity before the lockdown. The comparisons in yearly screening, i.e. comparing 2019 with 2020, should rather be broken down more in line with important dates of the epidemic.

We agree with the Reviewer that it would be much more informative to use exact dates; however, two limitations prevented us to do so. (1) the data have been provided to us on a weekly base. In principle, we could have used the weekly date to have a better time resolution. However, (2) when using weekly data, the seasonal decomposition procedure, used to isolate the normal seasonal variations (mainly due to holidays), failed to identify moving holidays (e.g. Easter) or holidays falling on different weeks in different municipalities (e.g. Winter vacation). These problems were solved by aggregating the data on a monthly base. We added the following to the limitations of the study (this include changes to account for question 1 of Reviewer #2):

“Finally, even though a number of tests have been carried out to check the validity of our analyses (i.e. analysis of (partial) autocorrelation functions, periodogram of Pearson residuals, non-randomized PIT histogram, marginal calibration plot, normality of residuals), our results should be further checked in different settings, such as different time series models and/or using data with higher time resolution (weekly, daily).”